# Value of Immunizations during the COVID-19 Emergency

**DOI:** 10.3390/ijerph18020778

**Published:** 2021-01-18

**Authors:** Armando Stefanati, Erica d’Anchera, Francesco De Motoli, Marta Savio, Maria Vittoria Toffoletto, Giovanni Gabutti

**Affiliations:** 1Department of Medical Sciences, University of Ferrara, 44121 Ferrara, Italy; gbtgnn@unife.it; 2Department of Medical Sciences, Postgraduate School of Hygiene and Preventive Medicine, University of Ferrara, 44121 Ferrara, Italy; dncrce@unife.it (E.d.); dmtfnc@unife.it (F.D.M.); svamrt@unife.it (M.S.); tffmvt@unife.it (M.V.T.)

**Keywords:** COVID-19, SARS-CoV-2, coronavirus, vaccines, immunization, public health, health policies, preventive strategies, infectious diseases, one health

## Abstract

Recent estimates by World Health Organization (WHO, Geneva, Switzerland) and United Nations International Children’s Emergency Fund (UNICEF) show a significant decline in vaccinal coverage rates in both pediatric and adult populations. The interruption of vaccination services is reported in at least 68 countries, with the involvement of about 80 million children worldwide. The situation is alarming if we consider that already in the period preceding the pandemic, immunization programs slowed down in various areas of the world. For these reasons, there is the risk of overloading health systems, already under pressure from the pandemic emergency, by employing human and economic resources for the management of epidemic outbreaks from vaccine-preventable diseases. The restoration and integration of vaccination services, the immunization of susceptible individuals as well as the adoption of adequate monitoring and surveillance measures are the main activities adopted by different countries to address the current global health emergency.

## 1. Aim of the Paper

The main aim of this paper is to describe the current situation in terms of vaccination planning and offer in the world after the outbreak of the SARS-Cov-2 pandemic, as well as what are the guidelines issued by the main world health organizations and the strategies globally adopted to cope with the decline in immunization following the slowdown in vaccination services. The manuscript speculates what can be done in the near future in order to minimize the possible consequences of the slowdown/suspension of vaccination services, establishing priorities for action and identifying the categories most at risk.

Another objective is to provide an overview of the Italian context, in particular on how the Prevention Departments have reacted and hence adapted to the pandemic and the restrictive measures consequently adopted to deal with it, pointing out which local contexts have suffered most from this situation and for what reasons. Further, the directives issued by the government and local authorities regarding the recovery and maintenance of the vaccination offer were evaluated.

Finally, we argue that, based on the evidence already present in the literature, there may be a possible beneficial impact of routine vaccinations on the outcome of the COVID-19 disease as well as in terms of reduced burden on the organizational and economic health system.

It should also be emphasized that the main limitation of this work is linked to the fact that the data available to date regarding the decline in vaccination coverage in Italy and in the rest of the world are still insufficient to draw conclusions on how much the pandemic has actually impacted preventive services. On the other hand, even if we still have partial information, there is a need to highlight the current situation, underlining the importance of being able to access the updated global data as quickly as possible, in such a way as to allow the reader to have an overview of how different nations are approaching the emerging issue of vaccination decline and the future global health consequences of a possible increase in vaccine-preventable diseases.

## 2. Pre-COVID-19 Vaccine Coverage Rates and Possible Impact of Vaccination Delay

Immunization against infectious agents responsible for potentially fatal diseases allowed to save millions of lives around the world. As a matter of fact, it is estimated that immunizations prevent 2–3 million deaths every year caused by diseases such as diphtheria, tetanus, pertussis, flu and measles. Immunization is therefore one of the largest and best investments ever made in terms of public health [1,2].

The COVID-19 pandemic, officially declared by the World Health Organization (WHO) on 11 March 2020, has severely impacted health systems around the world, particularly affecting basic health care services such as those dedicated to the immunization of the pediatric and adult population [3,4]. WHO, United Nations International Children’s Emergency Fund (UNICEF) and Global Alliance for Vaccines and Immunization (GAVI) report an interruption of routine immunization programs in at least 68 countries, which has involved around 80 million of children. In low-income countries supported by GAVI, 24 million people are at risk of missing out on vaccinations against measles, polio, rotavirus, meningitis, rubella and HPV. In the first five months of the COVID-19 pandemic, several countries suspended their immunization campaigns [5,6].

This will inevitably lead to an increased risk of vaccine-preventable diseases (VPDs) in the near future, increasing pressure on already greatly compromised health and social systems. Specifically, maternal and child health will be most affected by this problem: accordingly to recent studies, low-income countries will undergo an increase in morbidity and mortality due to the collapse of routine health services indirectly sustained by COVID-19 [3,7].

In the period prior to the pandemic, there had already been a global deadlock of vaccination coverage rates for DTP3 (three doses of diphtheria, tetanus and pertussis combined vaccine), which in 2019 stood at around 85%, as well as for measles. Based on available data, it is estimated that in 2019, about 14 million children did not receive any vaccine dose against these diseases. Accordingly to UNICEF estimates, approximately 19.7 million children worldwide have not received the three recommended doses of DTP before one year of age, and approximately 13.8 million children of the same age group have never been vaccinated [2]. According to a provisional estimate by WHO, in the period between January 2020 and April 2020, approximately 1.4 million fewer doses of DTP were administered in the world compared to the same period in 2019 [8].

In the United States, the evaluation of the January–April 2020 versus 2019 period showed that there was a 21.5% drop in the administration of the measles vaccine. Vaccinations against hepatitis, meningitis, polio and rotavirus also showed a decreasing trend compared to the previous year. Among the single US states that reported a decrease in vaccinations compared to 2019, immunizations suffered an overall decline of 63%, 40% and 45% in New York, California, Ohio and Virginia, respectively. In Virginia, for example, HPV vaccination had the greatest decline, equal to 65%. As for adults, a 30% reduction in the immunization rate was observed compared to May 2019, more evident in the 19-49 age group [6].

Countries such as those belonging to the southern region of Asia (particularly India, Nepal and Pakistan), which over the past decade have made enormous progress in terms of vaccination coverage, risk seeing jeopardized all the work done because of the pandemic [9,10]. In Indonesia, about 84% of immunization services have been blocked by the pandemic; 20% of these were related to measles and rubella vaccinations [6]. The situation in Latin America and the Caribbean could be potentially even more dramatic as vaccination coverage had already suffered a collapse during the last decade, and it will now suffer an even greater decline due to the COVID-19 pandemic [10].

Over the past decade, several countries have already faced tragic events that have compromised their immunization programs; some countries have managed to recover within a year, while others still struggle to restore their vaccination services. In this regard, in-depth studies on factors that have enabled countries to recover within a year of the crisis could help to quickly restart immunization programs while the COVID-19 pandemic is still ongoing and to not have a further decrease in immunization rates [8].

While global health is trying to reorganize in order to recover, UNICEF and WHO have supported nations to ensure that the objectives set in terms of coverage rates are again achieved, mainly by strengthening services and providing support to healthcare professionals [10].

## 3. Future Scenarios: Resilience, Reorganization of Services and Vaccination Offer

The WHO published an interim guide on March 26, 2020, with general guidelines for the management of vaccination activities during the COVID-19 pandemic. The interruption of immunization services, even for short periods, can result in an increase in susceptible individuals and consequently with an increase in epidemic outbreaks of VPDs (e.g., measles). The increase in the incidence of these diseases can test a health system already under pressure to manage the pandemic by increasing morbidity and mortality in groups most at risk, such as children and elderly. Given the high possibility of VPDs epidemics, it is of primary importance to ensure the continuity of the vaccination offer, guaranteeing the absolute safety both for healthcare professionals and for users [11,12,13,14].

Based on the experience gained during previous epidemics and health emergencies, support of essential health services such as vaccinations is a priority concern of the WHO. However, it should be considered that we are facing an unprecedented pandemic [12,13,14].

The WHO underlines that vaccinations are a fundamental health service and, as such, must be a priority and must be safeguarded even during the pandemic. The administration of vaccines must be conducted in conditions of complete safety, both for health professionals and for the general public. The surveillance of VPDs and any related outbreaks must be maintained and strengthened to allow for early diagnosis and case management. These measures could help to monitor the epidemiology of COVID-19. National authorities should constantly monitor the dynamics of COVID-19 in their respective countries and must play an important role in providing indications regarding maintenance, suspension and/or restoration of vaccination services. In the event of a decrease in vaccine delivery due to the COVID-19 pandemic, countries need to design effective strategies for planning recovery interventions. It is also recommended to temporarily suspend mass vaccination campaigns due to the risk of infection from SARS-CoV-2 or to carefully evaluate the risk-benefit ratio of a mass vaccination case by case.

The correct reprogramming of vaccination plans is based on an adequate and updated assessment of the epidemiological scenario of SARS-CoV-2 infection, differentiated by WHO in four possible scenarios: absence of cases, sporadic cases, cluster of active cases or presence of a high community transmission of the virus. WHO believes that every Ministry of Health must make all possible efforts to maintain high immunization rates in the population, also involving non-governmental organizations or civil associations to support vaccination programs in the most serious scenarios of SARS-CoV-2 spread in order to primarily recover the cohorts of children whose vaccination has been postponed and by implementing a “catch-up” vaccination recovery plan tailored to each risk group. It remains a priority to immunize infants with primary vaccinations, taking advantage of birth points, and to vaccinate the groups most at risk against pneumococcus and flu, postponing the introduction of new vaccines in national vaccination schedules.

Every effort must be tailored to the scenario in place in each country and must follow the eight pillars of preparation and response to a public health emergency identified by WHO, which provide for:a national coordination of monitoring and planning;a correct communication of the risk to the population;effective surveillance and active investigation of cases with rapid response;health checks at cross-border entry points;the presence of formally recognized national laboratories;the implementation of an infection prevention and control service through the timely updating of technical documents for activities related to prevention;the definition of therapeutic protocols for clinical case management;the implementation of operational and logistical support for all procedures for the acquisition of goods necessary for the management of the pandemic, such as diagnostic tests and medications, including vaccines.

The availability of vaccines must be assessed at a national level in order to possibly provide for the introduction of flexible timing between the administration of one booster and the other in order to temporarily meet the requests [12,13,15,16].

Regarding the recommendations for influenza vaccination for the year 2020, WHO underlines how the transmission of influenza viruses may have been affected by the preventive measures adopted to face the SARS-CoV-2 pandemic (e.g., social distancing, use of masks, etc.). The impact of these measures is supported by data from the southern hemisphere for the year 2020, where a substantial reduction in flu infections has been registered. On the other hand, the co-circulation of SARS-CoV-2 and influenza viruses can further impact health systems and the fragile population. For these reasons, WHO has deemed it appropriate to publish a supplementary document to the position paper on influenza dated 2012, confirming health workers, elderly, people with chronic diseases and pregnant women as groups at risk for which vaccination is recommended. In the document dated 21 September 2020, subjects at risk have been further classified, giving priority to healthcare workers and elderly over pregnant women, subjects at risk and children. Healthcare workers and the elderly are among the groups with the highest priority: the former must be vaccinated to avoid work absenteeism and to reduce the spread of viruses within health facilities, and the latter must be vaccinated as they are at a greater risk of severe disease both for COVID-19 and for flu-related complications. It is highlighted that residents in nursing homes and those >50 years of age should be identified as subjects to be urgently vaccinated, particularly in areas at greater risk for COVID-19. However, this new classification of at-risk groups should not negatively impact on already planned flu vaccination programs [17].

To date, the risk of having cohorts of subjects susceptible to vaccine-preventable diseases has significantly increased for several reasons; the on–off functioning of vaccination services and the logistical issues of health facilities related to the unavailability of health personnel engaged in the pandemic have heavily contributed to vaccination delays. At the same time, the preventive measures put in place to contain the SARS-CoV-2 pandemic have hindered the recipients of vaccinations to reach the immunization services. In addition, the fear of contracting the virus as well as travel restrictions have induced several subjects to voluntarily postpone already scheduled vaccinations.

According to the WHO, this risk related to decreasing immunization rates could be contained thanks to “catch-up” strategies providing for rescheduling of the vaccination offer for those who have missed the appointment. Among these strategies we can find the selective recall of the most at-risk cohorts (children, elderly, pregnant women, subjects at risk due to chronic diseases and subjects in areas affected by epidemic outbreaks of infectious diseases), the empowerment of institutions and local organizations and the implementation of human and environmental resources with adequate infrastructures. Together, these measures can make a difference in maintaining adequate vaccination coverage aimed at increasing herd immunity.

Another relevant factor is the control on how organizational changes are going on and the planning of new ones, if necessary. The correct respect of vaccination schedules in the foreseen ages is absolutely relevant as is the lessening of the risks of transmission of SARS-CoV-2. Taking into account the wide range of symptoms of COVID-19 in the population (from cases with clear symptoms to completely asymptomatic subjects), it is important to ensure compliance with preventive hygiene and behavioral rules against the spread of the virus within the vaccination services. In the case of an unvaccinated subject and/or an unknown vaccination history, healthcare workers must consider the subject as susceptible and provide them with vaccination adequate for their risk range and age.

To date, there is no contraindication to immunize subjects potentially positive for SARS-CoV-2 (asymptomatic) or who have been in contact with people with COVID-19. There is no evidence of an increased risk of contracting COVID-19 after a vaccination or that vaccination could influence the course of COVID-19 infection. Despite this, it is recommended to postpone the vaccination of subjects who tested positive for SARS-CoV-2 until the suspension of the precautions aimed at avoiding the spreading of the virus is envisaged for them.

Specifically, it is recommended to wait ten days from resolution of symptoms for symptomatic subjects and ten days from the test with positive result for asymptomatic subjects. Ultimately, both for contacts with positive SARS-CoV-2 cases and for subjects clinically cured from COVID-19, it is recommended to be vaccinated as soon as possible, in case one or more routine vaccinations have been missed. To ensure more effective immunization while limiting the opportunities for potential contagion, it is recommended to co-administer multiple vaccines and to use combined vaccines, if possible [18].

## 4. The Italian Context

The launch of research by the Ministry of Health has allowed us to obtain relevant information on how the COVID-19 emergency has impacted on routine vaccination activities throughout the Italian territory. Based on available data, it was found that vaccination activity has slowed down practically throughout the country, albeit unevenly in the various regions [19]. One of the reasons for this slowdown is linked to the relocation of health personnel and in particular of those involved in this service (more than 33% throughout the national territory) who have been called upon to support the management of the pandemic in the Public Health Departments. In addition to this, the measures of social isolation and distancing initially led to the reduction or even the suspension of vaccination activities. The latter situation in particular involved about a quarter of the country’s vaccination centers; this percentage has been even higher in the Lombardy region, an area that has suffered most from the effects of the epidemic due to the high number of people affected by the virus [20,21].

From a temporal point of view, the greatest decrease in coverage rates was observed in the period prior to the national lockdown announced on March 9, 2020, i.e., in conjunction with the notification of the first cases in Italy, and secondly when the epidemic reached its peak. In reality, the reduction in vaccinations did not affect the entire population in a uniform manner, but mainly concerned the pediatric group (from >1 year of age to adolescence) and to a lesser extent the adult population. This happened as, during the emergency context, an attempt was made to give priority to the most susceptible group, that is, that of infants under one year of age. Above all, the immunization with the greatest decline in coverage rate was that against HPV, normally offered to adolescents, followed by the one against Herpes Zoster, offered instead to the 65-year-old cohort. To a lesser extent, a decrease was also observed for DTaP and anti-meningococcus B immunizations [20,21].

In line with what has already been published by WHO, the Ministry of Health, through a circular dated 1st June 2020, has issued guidelines regarding the gradual reactivation of scheduled activities. Three categories of measures have been identified:access control and regulation of admissions: remote booking activity has been privileged, through booking centers, pharmacies or website;flow control: dedicated paths for users (with separation of entry and exit from the structure), wider opening hours and scheduled appointments in order to avoid overcrowding in waiting rooms;protocols and procedures relating to prevention and to protective and hygiene measures: need to implement preventive measures by measuring body temperature and any respiratory symptoms in subjects entering the services, correct use of personal protective equipment (PPE) and proper hand sanitation. It has been also important to carry out constant sanitation and ventilation of the environments and to adequately train the staff on COVID-19 (Table 1) [20,21].

Emphasis has also placed on the centrality of communication with the population and in particular on the importance of making families understand how necessary it is to continue to carry out vaccinations and how these represent effective prevention measures together with those adopted to deal with SARS-CoV-2 [22]. In this sense, rescheduling of appointments has certainly increased the opportunities for contact between users and healthcare providers and for exchanging information and solving doubts. Promotion of vaccinations has involved, in addition to the staff of the vaccination services themselves, other professionals such as general practitioners, pediatricians and other specialists. It has been possible to apply some flexibility in the vaccination schedule to recover unvaccinated subjects, applying the most appropriate protocol based on the available scientific evidence. There has also been an urgent need to expand routine vaccination services through the search for alternative spaces, always in compliance with anti-COVID-19 regulations and by strengthening dedicated staff, also through a progressive autonomy of non-medical staff in the management of vaccination sessions [20,21,23,24].

One of the crucial points to allow the effective and continuous work of vaccination centers is to find new organizational methods that can support ordinary activities but above all that bring substantial and ready help during emergency situations that no longer seem so rare. Considering the proportion of individuals who have not received the planned vaccinations since the beginning of the pandemic, it is very likely that the overwork due to the accumulation of vaccination sessions will result in another type of emergency that the Public Health Departments will face in a not so far future. In addition, the future availability of the SARS-CoV-2 vaccine will lead to a further change in the organization of the vaccination centers. The introduction of this new vaccine should not distract attention from the remaining programming, which must absolutely resume at full capacity, recovering all canceled or missed appointments as soon as possible. Given the rapidity with which today’s scenarios change, it is essential to train personnel who can help to achieve this goal. It is desirable that with the same promptness with which doctors, nurses and health care assistants have been relocated to deal with the pandemic, the activities of health personnel will be organized to recover the vaccination sessions that have remained pending. To this end, one of the organizational strategies that is already being implemented in Italy for the flu campaign is the involvement of pediatricians, general practitioners and other specialists in promoting and administering vaccines. This is a winning choice not only in times of emergency such as the one we are currently experiencing but also in a future perspective where the anti-SARS-CoV-2 vaccine will arrive and will have to be administered to the population. Given the reasons that led citizens to give up vaccination sessions for fear of compromising their safety, it is important to work more on the relationship of trust within which vaccinations should be carried out. For this purpose, the role of trusted doctors (pediatricians, general practitioners and specialists who follow the chronic patient) is once again central, as they are figures constantly present in the lives of citizens and have the best chance of establishing a relationship that makes the individual feel safe even in the most critical moments. Obviously, everything must be developed jointly with the Public Health Departments so that the aforementioned doctors have all the necessary tools to collaborate actively and effectively [25,26].

That first big test is the 2020–2021 flu season. The law decree of 14 August 2020 already ordered the regions to submit within a month (therefore by mid-September 2020) an organizational plan for the recovery and management of the waiting lists created during the first months of the pandemic. Despite the presence of a national legislative obligation, there is a strong lack of homogeneity in the response of different regions. Currently, not all the regions have drawn up their plans, and this is already indicative of how much the Health System is under pressure. In many Italian regions, the flu campaign started earlier than in previous years in order to have an early and timely immunization. In some regions (Liguria, Lazio and Abruzzo) the free-of-charge vaccination has also been extended to the age group from 6 months to 6 years. In all the regions that drafted a plan for the flu campaign, the vaccine doses forecasted were 20–50% higher than the previous year [27].

For example, the Veneto region plan for autumn 2020 provides for specific interventions depending on the sectors involved. The various areas of intervention include the aforementioned prevention departments envisaging a strengthening of the pneumococcal and flu vaccination campaigns. Pneumococcal vaccination campaign for nursing homes has already been launched in the region and is expected to be completed; to this is added the annual campaign for 65-year-olds, which now provides for an extension of the free-of-charge offer to the cohort of subjects born since 1943. As regards flu vaccination, the campaign was anticipated in October as well as an enhancement of virological surveillance for Influenza and SARS-CoV-2. For the implementation of both vaccination campaigns, the support of general practitioners (GPs), pediatricians (PLS) and special care continuity units (USCA) was requested [28].

## 5. Possible Correlation between Vaccinations and COVID-19: Recent Studies

Scientific research is considering the possible relationships between already available vaccinations and SARS-CoV-2 infection. Although there is currently a lack of data on how influenza vaccination could possibly impact SARS-CoV-2 infection and COVID-19 severity, recent studies have nonetheless examined the possible relationship between infection with certain strains of coronavirus already present before the current pandemic and seasonal flu vaccination. Preliminary data seem to support that vaccination may, in some way, have a favorable effect on COVID-19 and its course, although the reason is not yet known [29].

Nevertheless, in consideration of the variable efficacy of flu vaccine and the high number of serotypes involved in pneumococcal infections not necessarily included in the pneumococcal vaccines recommended for adults, a possible but not particularly incisive impact of these two vaccinations on morbidity and mortality from COVID-19 is hypothesized. Furthermore, the presence of infection or carriage of specific pathogenic bacteria has been evaluated in patients with COVID-19 and non-COVID-19 pneumonia. *Bordetella pertussis* was statistically more frequent in patients with COVID-19 pneumonia, and this suggests that *B. pertussis* should be considered more carefully in the treatment of COVID-19, highlighting the importance of pertussis vaccination as well [30].

Other researchers propose a theory of testing the *Bordetella pertussis* vaccine for protection against COVID-19 in adults [31]. Others argue that whooping cough is a vaccine-preventable infectious disease of the airways and shares many similarities with COVID-19, including transmission mode and clinical features. Although pertussis is caused by a bacterium (*Bordetella pertussis*) while COVID-19 is a viral infection (SARS-CoV-2), some data seem to support that cross-reactivity and heterologous adaptive responses can also be observed among unrelated agents such as between bacteria and viruses [32].

In general, a possible indirect effect of vaccinations could be to decrease the burden of hospitalized patients and at the same time decrease the risk of health-care-related COVID-19 infection. It remains therefore of fundamental importance to have reliable data on the incidence of all these diseases in order to verify these hypotheses [33].

## 6. Conclusions

The COVID-19 pandemic has placed particular emphasis on the value of collecting high-quality data needed to monitor immunization programs and to stimulate all countries to account for their progress. The availability of valid and equitable immunization programs is essential to avoid vaccine-preventable disease outbreaks. In times of emergency such as we are, it is desirable to have a forward-looking vision on the possible future scenarios.

As regards the ethical aspect of administering already available vaccines and in view of the arrival of a vaccine against SARS-CoV-2, it is appropriate to act in terms of transparency towards public opinion, making the population aware of guidelines for the rational use of available resources. With a view to the arrival of the vaccine against SARS-CoV-2 it is also necessary to re-evaluate the approach to vaccination hesitancy, which despite the pandemic is growing also due to the currently circulating infodemic and the “fake news”. The strengthening of the “mandatory vaccination” policy may not necessarily lead to the desired results if the population is not adequately educated and updated on the aims of an effective vaccination coverage for VPDs both at national and European level.

To identify the priority groups to which administer the SARS-CoV-2 vaccine, it will be appropriate to consider first the groups with the worst outcomes (such as the elderly), healthcare personnel, nursing-homes residents, people suffering from chronic diseases (diabetes, cardiovascular disease, hypertension, obesity, etc.), essential workers exposed to greater risk (such as transport workers and enforcement authorities) and finally the population most at risk of not receiving adequate care (poor and fragile subjects). Thismust be done while taking into account that the principle of “first come, first served” is not applicable in emergency conditions, in the same way the criterion of “everything to everyone” applied to the vaccine against COVID-19 can be applied only when the aforementioned risk groups will be protected [34,35,36,37]. This paper will be helpful in planning future vaccination campaigns and in stimulating deep reflection on the importance of this part of public health that today more than ever has a significant impact on the future of our lives.

## Figures and Tables

**Table 1 ijerph-18-00778-t001:** Organizational strategies for the safety of immunization services. Modified from [20,21].

Access Control and Regulation	Definition of Separate Routes and Flow Controls	Protocols and Procedures Relating to Prevention and Protection and Hygiene Measures
Remote booking	Paths dedicated to users	Body temperature measurement
Booking through reservation centers or pharmacies	More flexible hours	Evaluation of respiratory symptoms or ILI *
Booking through the company website	Staggered appointments	Proper hand hygiene, sanitation and ventilation of environments

ILI *: Influenza-like illness that includes myalgia, arthralgia, fever, chills, fatigue, nausea, loss of appetite or bone pain.

## Data Availability

No new data were created or analyzed in this study. Data sharing is not applicable to this article.

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
