# Peer review of "Value of Immunizations during the COVID-19 Emergency"

_ijerph, 2021, doi:10.3390/ijerph18020778_

Round 1
Reviewer 1 Report
REVIEW IJERPH_30-11-2020
This paper is very interesting, but it cannot be published at the moment and needs to undergo some changes.
Some suggestions are the following:
- What is the scope/aim and what are the objectives of this paper?
- Provide with specific objectives that you are in following addressing, (otherwise it is just another plain descriptive paper and we do not need to read this one).
- Why is this paper important in academia?
- Who is it important for?
- Why has it been written?
- Who and how will benefit from this paper?
- Has this issue been addressed before? If not this is a gap in academia that you are trying to close
- Provide with an introduction in this paper that addresses and builds the above and then analyze per objective.
- There are a number of factors that could be the reason why people or countries just do not want to do vaccinations, which are not mentioned such as distrust to the vaccines, allergies, money cut offs in the health sector etc.
- Is the focus on Italy? If yes provide it as an objective.
- Also, maybe n.4 line 281-307 should go before Italy.
- The way the paper is handled – even though in very good English, and what is written is important, one does not know why he/she is reading it.
- For this to be overcome, provide with an introduction that will build your case that vaccinations are even more significant in the covid-19 era bt there are declines in vaccinations…., and for so, this paper has as its aim to describe…. (if that is what you want to do) and its objectives are….. (Italy as a case study?)
- Based on these unfold your paper by aim and objective in order to be clear.
- What is the final point that you want to make? It has to come out from what you present. Since you do not have any type of research (qualitative or quantitative) you need to have a strong presentation. Otherwise it is just a descriptive paper, that does not bring something new to the table, except that vaccinations are important (and we already know that).
16. Provide more analysis and depth in conclusion, and provide with limitations of study, as well as implications
17. Conclude to the importance of your study for academia
I hope that these comments will help you with your work
Best of luck with your paper and be safe!
Author Response
This paper is very interesting, but it cannot be published at the moment and needs to undergo some changes.
Some suggestions are the following:
What is the scope/aim and what are the objectives of this paper?
Thank you for your question: we added a new paragraph in the first part of the document.
Provide with specific objectives that you are in following addressing, (otherwise it is just another plain descriptive paper and we do not need to read this one).
Thank you for pointing this out. We added the specific objectives in the first paragraph.
Why is this paper important in academia?
Who is it important for?
Why has it been written?
Who and how will benefit from this paper?
We answered all these questions in the first part of the document, thank you.
Has this issue been addressed before? If not this is a gap in academia that you are trying to close
Thank you for your question. We think that this topic has not yet been fully addressed in the literature and we have decided to summarize the international guidelines and provide our opinion on the matter.
Provide with an introduction in this paper that addresses and builds the above and then analyze per objective.
We have added this in the first part of the document, thank you.
There are a number of factors that could be the reason why people or countries just do not want to do vaccinations, which are not mentioned such as distrust to the vaccines, allergies, money cut offs in the health sector etc.
Thank you for pointing this out. The purpose of the document is to focus attention on the vaccination decline and its future consequences. Unfortunately, fears about vaccinations in general have been exacerbated by the pandemic and it is therefore even more important to focus on maintaining adequate vaccination coverage, referring to the various guidelines and implementation methods that we have mentioned.
Is the focus on Italy? If yes provide it as an objective.
Thank you, done.
Also, maybe n.4 line 281-307 should go before Italy.
Thank you for pointing this out. We would prefer to keep the original paragraph order.
The way the paper is handled – even though in very good English, and what is written is important, one does not know why he/she is reading it.
Thank you for pointing this out. We think we provided a new perspective on this topic.
For this to be overcome, provide with an introduction that will build your case that vaccinations are even more significant in the covid-19 era bt there are declines in vaccinations…., and for so, this paper has as its aim to describe…. (if that is what you want to do) and its objectives are….. (Italy as a case study?)
Thank you for pointing this out. We added the objectives in the first paragraph.
Based on these unfold your paper by aim and objective in order to be clear.
Thank you for your suggestion, we added a new part in the paper to be clearer.
What is the final point that you want to make? It has to come out from what you present. Since you do not have any type of research (qualitative or quantitative) you need to have a strong presentation. Otherwise it is just a descriptive paper, that does not bring something new to the table, except that vaccinations are important (and we already know that).
We think that is important to focus on new problems that are emerging from COVID-19 pandemic, such as the lack of data on vaccinations coverage. We added a new part in the presentation, thank you for the suggestions.
Provide more analysis and depth in conclusion, and provide with limitations of study, as well as implications
Thank you, done.
Conclude to the importance of your study for academia
Thanks for your suggestion, we have implemented the conclusion paragraph.
Reviewer 2 Report
Many thanks for giving me the opportunity to be the first reader of the review entitled “Value of immunizations during the COVID-19 2 emergency”. Although I am a statistician, I read this article with great care because it contains a lot of interesting material about the state of vaccination in the current pandemic, and this has increased my knowledge of COVID-19. I think other readers would get the same impression if this article was printed.
This manuscript is a scientific review, although it does not follow standard parts: introduction, aims, methods, results, discussion, etc. It seems like the essay. There are five chapters that logically develop the idea of the need for immunization. The authors draw on a large amount of material describing the situation around the world: the United States, East Asia, Latin America. All sources of information are published in 2020!
Below are some my comments:
- Despite all the above advantages, it should be written at the beginning of the article for what purpose this review was prepared and on what material it is based.
- The structure of the article is monotonous, a bit boring to read. Therefore, it should be more structured (e.g. lines 120-130), to highlight the main ideas in the individual boxes.
- It is likely that this publication will become the basic document on which health policy will be based. So, shouldn’t practical recommendations be given to doctors and health policy makers?
Finally, I would like to congratulate the authors for the work presented.
Author Response
Many thanks for giving me the opportunity to be the first reader of the review entitled “Value of immunizations during the COVID-19 2 emergency”. Although I am a statistician, I read this article with great care because it contains a lot of interesting material about the state of vaccination in the current pandemic, and this has increased my knowledge of COVID-19. I think other readers would get the same impression if this article was printed.
This manuscript is a scientific review, although it does not follow standard parts: introduction, aims, methods, results, discussion, etc. It seems like the essay. There are five chapters that logically develop the idea of the need for immunization. The authors draw on a large amount of material describing the situation around the world: the United States, East Asia, Latin America. All sources of information are published in 2020!
Below are some my comments:
Despite all the above advantages, it should be written at the beginning of the article for what purpose this review was prepared and on what material it is based.
Thank you for pointing this out. We added in the first part a description of our objectives.
The structure of the article is monotonous, a bit boring to read. Therefore, it should be more structured (e.g. lines 120-130), to highlight the main ideas in the individual boxes.
Thank you for your suggestion. We have modified the parts underlined in green, for easier reading.
It is likely that this publication will become the basic document on which health policy will be based. So, shouldn’t practical recommendations be given to doctors and health policy makers?
Thank you for pointing this out. We hope that the local health authorities will take inspiration from our paper to develop guidelines specific to the epidemiological situation of their country.
Reviewer 3 Report
Title: Value of immunizations during the COVID-19 emergency
This article emphasizes the impact of the coronavirus epidemic on the resources allocated to the vaccination of the most preventable infectious diseases. The emergence of these diseases can place an additional burden on health systems, especially in disadvantaged countries. The different regional impact on conventional vaccination is noteworthy. It contains strategies to reduce the harm to the population from delayed compliance with the vaccination program, especially in children.
I think it is a necessary article for corrective measures to be taken for the vaccination deficit we are suffering from. The work is divided into several sections that rightly analyze the different aspects of this problem: vaccine coverage rates before COVID-19 and the possible impact of the delay in vaccination, future scenarios: resistance, reorganization of services and supply of vaccination, the Italian context and possible correlation between vaccines and COVID-19: recent studies.
Author Response
This article emphasizes the impact of the coronavirus epidemic on the resources allocated to the vaccination of the most preventable infectious diseases. The emergence of these diseases can place an additional burden on health systems, especially in disadvantaged countries. The different regional impact on conventional vaccination is noteworthy. It contains strategies to reduce the harm to the population from delayed compliance with the vaccination program, especially in children.
I think it is a necessary article for corrective measures to be taken for the vaccination deficit we are suffering from. The work is divided into several sections that rightly analyze the different aspects of this problem: vaccine coverage rates before COVID-19 and the possible impact of the delay in vaccination, future scenarios: resistance, reorganization of services and supply of vaccination, the Italian context and possible correlation between vaccines and COVID-19: recent studies.
Thanks for your positive comments.
Reviewer 4 Report
The manuscript deals with a very current problem but does not give new information and does not provide a new scientific contribution. The paper is written in a simple way with little clinical impact. Literature research is very scarce and should be improved. I do not find it suitable for a scientific publicationAuthor Response
The manuscript deals with a very current problem but does not give new information and does not provide a new scientific contribution. The paper is written in a simple way with little clinical impact. Literature research is very scarce and should be improved. I do not find it suitable for a scientific publication
Thank you for your opinion. We are sorry that you didn't like it, we hope the new additions may have improved our article.
Round 2
Reviewer 1 Report
dear authors
this paper has been improved.
only two points:
1. line 43 : "Finally, we wanted to analyse: please rephrase to "Finally, we analyzed...."
2. line 373 :"We hope that this paper will be helpful in" please rephrase to (or something like that): " this paper, taken to account, will be helpful..."
I hope these suggestions will help your work
Happy new year & Be safe!
Author Response
Dear authors, this paper has been improved. Only two points:
1. line 43 : "Finally, we wanted to analyse: please rephrase to "Finally, we analyzed...."
2. line 373 :"We hope that this paper will be helpful in" please rephrase to (or something like that): " this paper, taken to account, will be helpful..."
I hope these suggestions will help your work
Happy new year & Be safe!
Thank you for your suggestion. We have used the track changes function to modify our work:
1) In line 43 we rephrase to "Finally we analyzed that based on the evidence already present in the literature, there may be a possible beneficial impact of routine vaccinations on the outcome of the COVID-19 disease as well as in terms of reduced burden on the organizational and economic health system."
2) In line 373 we rephrase to "This paper, taken to account, will be helpful in planning future vaccination campaigns and in stimulating deep reflection on the importance of this part of public health that today more than ever has a significant impact on the future of our lives."
We thank you very much for your comments that allowed us to improve our work.
Reviewer 4 Report
I appreciated the changes made to the paper I think it can be published without further revisions.Author Response
Thank you very much for the positive comment, we are glad that we were able to improve our paper.